# Overfeeding-Induced Obesity Could Cause Potential Immuno-Physiological Disorders in Rainbow Trout (*Oncorhynchus mykiss*)

**DOI:** 10.3390/ani10091499

**Published:** 2020-08-25

**Authors:** HyeongJin Roh, Jiyeon Park, Ahran Kim, Nameun Kim, Yoonhang Lee, Bo Seong Kim, Jasna Vijayan, Mu Kun Lee, Chan-Il Park, Do-Hyung Kim

**Affiliations:** 1Department of Aquatic Life Medicine, College of Fisheries Science, Pukyong National University, Busan 48513, Korea; hjroh@pukyong.ac.kr (H.R.); jiyeon1388@naver.com (J.P.); ahran110@naver.com (A.K.); skansl123@naver.com (N.K.); dldbsgkd07@naver.com (Y.L.); jasnavijayan@gmail.com (J.V.); 2Department of Chemistry, Center for Proteome Biophysics, Chemistry Institute for Functional Materials, Pusan National University, Busan 46241, Korea; 3Aquatic Disease Control Division, National Institute of Fisheries Science (NIFS), Busan 46083, Korea; fishpath@korea.kr; 4Korean Aquatic Organism Disease Inspector Association, Busan 46008, Korea; leemukun@naver.com; 5Department of Marine Biology & Aquaculture, College of Marine Science, Gyeongsang National University, Tongyeong 53064, Korea

**Keywords:** overfeeding-induced obesity, OxLDL, immune disorder, rainbow trout, apoptosis, nutritional disease

## Abstract

**Simple Summary:**

In this study, we have successfully generated overfeeding-induced obesity in rainbow trout and demonstrated that the overfeeding regime causes adverse effects on the health status of fish. What we found in this study is that fish in the overfed group harbor enlarged liver and macrophages due to high lipid accumulation. The expression of IL-10, CD36, TLR2, and HSP70 (stress and/or obesity-related genes) was significantly upregulated in overfed fish. Moreover, oxidized low-density lipoprotein (OxLDL), which is known to be produced more in obese individuals, caused apoptosis of trout lymphocyte. These results clearly indicate that overfeeding-induced obesity can be the source of stress and cause immuno-physiological disorders in rainbow trout. Awareness and knowledge of farmers on the relationship between overfeeding and fish health might contribute to increased disease resistance and aquaculture production.

**Abstract:**

Although over-nutrition from overfeeding-induced obesity is known to be highly associated with metabolic and immunological disorders in humans, little is known about overfeeding-induced obesity in fish farming. The purpose of this study was to investigate changes in immuno-physiological parameters, to better understand the potential risk of overfeeding–induced obesity in fish. Commercial feed was provided to fish in the overfed group until they refuse to eat, but fish in the control group was fed with the feed at 1% bodyweight per day. The hemato-serological, histological, and immunological changes were observed at weeks 2 and 8. Rainbow trout leukocytes were co-incubated with oxidized low-density lipoprotein (OxLDL), and the phagocytes engulfing the OxLDL and the presence of apoptotic cells were evaluated. The body weight, body mass index (BMI), and hepatosomatic index (HSI) index were significantly higher in the overfed group, and high lipid accumulation and fatty changes were also observed in their livers, indicating that the feeding regime used in this study led to overfeeding-induced obesity. Likewise, much higher numbers of and larger vacuoles were observed in overfed fish macrophages, showing unclear boundaries between the cytoplasm and extracellular space. In the overfed group, the expression of IL-10, HSP70, TLR2, and CD36 was significantly higher, and lymphocyte apoptosis was more evident, indicating that overfeeding-induced obese fish might have immunologic disorders. This was the first study to demonstrate that overfeeding-induced obesity could cause an immune-physiological imbalance in rainbow trout, making them more vulnerable to infectious diseases and various stressful conditions. This study will contribute to improvements in fish nutrition, feeding practices, fish nutrition, and disease prevention in the aquaculture industry.

## 1. Introduction

Nutrient balance is one of the most essential factors affecting the health status of farmed fish [1]. Several studies have emphasized that optimizing the feed quantity is very important for farmed fish as overfeeding is one of the major stress sources and might lead to nutritional imbalance [1,2,3,4]. Although there are no statistics regarding overfeeding in fish farms, it might be common due to the use of a poor feeding strategy [5]. Excess nutrients not utilized in fish farms are easily released into the environment and make secondary pollution [5]. Inappropriate feeding strategies that result in overfeeding decrease feed efficiency, and increase feed waste, bacterial loads, and environmental degradation [6,7,8]. Overfeeding in fish farms seems common, but its risk to fish has not been well clarified.

In humans, surfeit and obesity cause medical problems, such as metabolic disorders, immune disorders, and diseases (e.g., atherosclerosis, hypertension, and diabetes) [9,10,11]. Lipid accumulation, especially atherogenic lipids, in humans can increase oxidized low-density lipoprotein (OxLDL) levels, which is highly related to metabolic dysregulation, chronic inflammation, and apoptosis in leukocytes [12]. LDL concentration in the blood of obese subjects is significantly increased, and LDL is highly susceptible to oxidization caused by oxidative stress [13,14]. Obesity causes leukocytes to secrete pro-inflammatory cytokines and may increase co-morbidities due to obesity [12,15]. Thus, the effect of OxLDL on the function of leukocytes is crucial in understanding the immune response in overfed fish. In recent years, Landgraf et al. [3] demonstrated that zebrafish overfed with a high-fat diet for eight weeks showed metabolically distinct obesity phenotypes, implying that fish could be used as a model to study the regulatory mechanisms for the development of metabolically healthy and unhealthy obesity. Cao et al. [16] found that fish fed a high-fat diet had excessive lipid levels in their livers, causing endoplasmic reticulum stress and ultrastructural impairment of the mitochondria, thereby inducing fatty liver disease. Likewise, immune responses, such as Mx, INF-*γ*, IL-6, IgM, CD8, CK11, TLR2, and TLR22 in adipose tissues, were hyperactivated when the high-fat feed was administrated to rainbow trout (*Oncorhynchus mykiss*) for a month [17].

Most studies on obesity in fish have focused on the effect of high-fat-induced obesity in different animal models. For example, health status and physiological changes were investigated in yellow croaker, goldfish, and grass carp after feeding high- and low-fat diets for several weeks [18,19,20]. Although the high-fat feed used in previous studies to investigate the influence of excessive fat and lipid, the artificial high lipid feed is not usually used in the fish farm [16,18,19]. However, overfeeding is common. Thus, understanding the immuno-physiological responses caused by overfeeding-induced obesity is necessary. Therefore, the purposes of this study were to investigate potential immuno-physiological disorders by profiling the changes in the hemato-serological, histological, and immunological parameters in overfed-stressed rainbow trout and examine the adverse effects of OxLDL, a major by-product of overfeeding-induced obesity, on trout leukocytes.

## 2. Materials and Methods

Ethical approval: The animal experiment was reviewed and approved by the Animal Research Ethics Committee at Pukyong National University (approval number: 2017-11).

### 2.1. Fish

Rainbow trout (bodyweight = 88.6 ± 14.0 g) were acclimated in continuously aerated freshwater at 15 °C. The fish were acclimated in recirculated 500 L of the oviform tank. Water was pumped into the acclimated tank in a clockwise direction, and 75% of the rearing water was exchanged daily. Commercial floating dry pellets (DongA One, Busan, Korea) consisting of crude protein (52%), crude ash (15%), crude fat (10%), phosphorus (2.7%), crude fiber (2%), and calcium (1.2%), were provided at 1% of the fish bodyweight for seven days.

### 2.2. Overfeeding Stress

Groups of 30 rainbow trout were acclimated in the control and overfed tanks. Fish in the overfed group were fed commercial floating feed daily until they refused to eat more at least three times a day by hand, while fish in the control group were fed with only 1% of their body weight (BW). Five fish in the control and overfed groups were sacrificed at 2 and 8 weeks after overfeeding (wao). The uneaten feeds were collected within 30 min after providing to trout. Peripheral blood was collected from the caudal vein after anesthesia using MS-222 (Sigma, Ronkonkoma, NY, USA), and was then treated with anticoagulant (Na-heparin or 3.8% sodium citrate) for further analyses. The head kidney was isolated and divided into two parts for RNA extraction and histology. The liver weight was measured to calculate the hepatosomatic index (HSI) by dividing the liver weight into body weight, and the parts were fixed in 10% neutral formalin for histological analysis. The body mass index (BMI) was calculated by the dividing weight (g) by the length squared (cm^2^).

### 2.3. Hematology and Serology

Hematological parameters, such as red blood cell (RBC) counts, hematocrit (Ht), hemoglobin (Hb), erythrocyte sedimentation rate (ESR), mean corpuscular volume (MCV, μm^2^), mean corpuscular hemoglobin (MCH, pg), and mean corpuscular hemoglobin concentration (MCHC) were calculated. Briefly, MCV was obtained using the following formula: MCV = 10 × Ht (%) divided by RBC count (in million units). MCH was calculated by dividing the total Hb concentration (g/dL) by the RBC count (in million units): MCH = ((Hb × 10)/RBC count). MCHC was used to measure the concentration of hemoglobin, and was calculated by the following equation: MCHC = (Hb /Ht) × 100. Blood samples were centrifuged at 12,000 rpm for 10 min to separate the plasma, and the serological parameters were measured using a dry chemistry analyzer as following manufacturer’s methods (FUJI DRI-CHEM 3000). The parameters analyzed were glutamic oxaloacetic transaminase (GOT), glutamic pyruvic transaminase (GPT), glucose (GLU), alkaline phosphatase (ALP), lactate dehydrogenase (LDH), total protein (TP), calcium (Ca), and total cholesterol (TCHO).

### 2.4. Histological Analysis

The liver and head kidney were fixed in 10% neutral-buffered formalin (BBC Biochemical, Mt Vernon, WA, USA), following a previous method [21]. The tissue sections of the liver and kidney were stained with hematoxylin and eosin (H & E) (BBC Biochemical, USA). The size of the macrophages was determined in the head kidney (area; μm^2^), and the cytoplasmic and vacuolated area were quantified in the hepatocytes using image analysis software (Computer program Image-Pro^®^ plus v.4.1, Media Cybernetics, Silver Spring, MD, USA).

### 2.5. RNA Extraction, cDNA Synthesis, and Real-Time PCR (Quantitative PCR)

Total RNA in the head kidney was extracted using the Trizol^®^ method (Life Technologies™, Carlsbad, CA, USA). The isolated RNA was treated with DNase I at a final concentration of 200-unit/mL (Sigma) and quantified using a NanoVue^plus^ (Biochrom US Inc., Holliston, MA, USA). One microgram of total RNA was used for synthesizing cDNA using M-MLV Reverse Transcriptase (Bioneer, Korea), and real-time PCR was performed to evaluate the mRNA expression level of cytokines, such as interleukin-1β (IL-1β), interleukin-10 (IL-10), chemokine-1 (CK-1), and tumor necrosis factor-α (TNF-α) [22], toll-like receptor (TLR)2, cluster of differentiation 36 (CD36), serum amyloid A (SAA), cluster of differentiation 3 (CD3), heat shock protein 70 (HSP70), and secreted type of IgM (sIgM). The expression of elongation factor 1 (Ef-1α) was used as an internal control, and fold-changes were calculated using the 2^–ΔΔCt^ method.

### 2.6. Leukocyte Isolation and OxLDL Treatment

Trout leukocytes were isolated from the head kidney using the Percoll method [23]. The leukocytes (~5 × 10^5^) in 1 mL of L-15 + 10% fetal bovine serum (FBS) + 1% antibiotic-antimycotic (Anti-Anti, 10,000 units/mL of penicillin, 10,000 μg/mL of streptomycin, and 25 μg/mL of amphotericin B; Gibco, Gaithersburg, MD, USA) were seeded into a 24 well microplate and incubated for 2 h at 15 °C. Then, the wells were treated with oxidized low-density lipoprotein (OxLDL, 37.5 μg). Likewise, the same volume of L-15 was inoculated for the control, and all the samples were assayed in triplicate to validate the data. Two to six hours post-inoculation (hpi), the leukocytes were detached by a cell scrapper and washed twice with added L-15 + 2% FBS + 1% Anti-Anti, and centrifuged at 200 g for 15 min at 4 °C. The leukocytes were then stained with Nile Red solution (1 μg/mL) (Enzo Life Sciences, Lausen, Switzerland) at room temperature (RT) for 10 min, followed by washing the cells twice. Ten-thousand leukocytes were randomly selected, and flow-cytometry analysis was performed to quantify Nile-red fluorescence using the FL-3 channel (Accuri C6™ Flow cytometer, BD Biosciences, San Jose, CA, USA).

### 2.7. Annexin-V/PI Double Staining Assay

Trout leukocytes (~5 × 10^5^ cells) were seeded in 12-microwell plates, and co-cultured in 1 mL of L-15 + 10% FBS + 1% Anti-Anti (Gibco, Gaithersburg, MD, USA) with 0, 0.2, 1, and 2 mg/mL of UV-OxLDL at 15 °C. The leukocytes without OxLDL treatment were used as controls. After 2, 8, 24, and 48 hpi, the leukocytes were detached using a cell scraper and centrifuged at 200 g for 10 min at 4 °C. After removing the supernatant, the leukocytes were double-stained with 10 μg/mL of propidium iodide (PI; Sigma) and 0.5 μg/mL of Alexa Fluor^®^ 647-conjugated annexin-V (BioLegend, San Diego, CA, USA) in 100 μL L-15 + 10% FBS + 1% Anti-Anti for 15 min at RT. After adding 400 μL of annexin-V binding buffer (0.01 M HEPES, 0.14 M NaCl, and 2.5 mM CaCl_2_), the fluorescence of PI and Alexa Fluor^®^ 647 in each cell was quantified using flow cytometry (Accuri C6™ Flow Cytometer, BD Biosciences, San Jose, CA, USA).

### 2.8. Statistical Analysis

The results presented in this study are expressed as mean ± standard deviation (SD), and subjected to one-way analysis of variance (ANOVA) in SPSS (17.0) based on Duncan’s multiple range test for BMI index. The Student’s *t*-test was carried out for serological and histological results, HSI, gene expression, and flow cytometry analysis. Significant differences between the groups were indicated by *p*-values of less than 0.05.

## 3. Results

### 3.1. Fish Feeding Rate and Growth

During the experiment, no mortality occurred in either groups. The control and overfed groups were fed commercial feed at 1% and 3.44 ± 0.57% of their bodyweight/day, respectively (Figure 1A). The mean bodyweight of the rainbow trout (initial mean bodyweight approximately 87 ± 14 g) and BMI significantly increased to 107 ± 27 g (BMI = 0.24 ± 0.02) and 195 ± 25 g (BMI = 0.29 ± 0.04) in the control and overfed trout, respectively, at week 8. Likewise, at 8 weeks, the overfed trout had significantly higher BMIs than those of the controls 2 and 8 weeks (Figure 1B).

### 3.2. Hematology and Hepatosomatic Index (HSI)

Table 1 presents the hemato-serological results (RBC counts, hematocrit, hemoglobin, MCV, MHC, MCHC, TP, ALP, GOT, GPT, GLU, TCHO, LDH, and Ca). Of those parameters, only the TP, hematocrit, and ALP values of the overfed fish were significantly higher than those of the fish in the control group. Specifically, the hematocrit and ALP of the overfed group were significantly higher from the hematocrit and ALP of the control group only at 8 weeks. The TP levels of the control at 2 and 8 weeks were 4.5 ± 0.4 and 4.9 ± 0.4 g/dL, respectively, whereas the TP levels in overfed fish at 2 and 8 weeks were significantly higher than the controls. Likewise, the HSI of overfed trout at 2 weeks and 8 weeks was significantly higher than that of the control (Table 1).

### 3.3. Histopathology of the Liver and Head Kidney

The histological results in the liver during overfeeding stress are shown in Figure 2, and different types of fatty changes were observed at 2 and 8 weeks. At 2 weeks, the overfed trout had large vacuoles in hypertrophic hepatocytes that were considered macro-vesicular fatty changes. They also had larger vacuolated areas (VA) and cytoplasmic area (Cya) compared to the control. In contrast, the fish with 8 weeks of overfeeding stress had several small vacuoles in their hepatocytes, and only the Cya, not the VA, was larger than that of the controls (Figure 2). Likewise, there were morphological changes in the head kidney leukocytes, including macrophages, of the fish in the overfed group. The boundary between the cytoplasm and extracellular space in the macrophage of fish in the control group was much clearer than that in overfed fish (Figure 3). The total cell area (Cy), Cya, VA, and vacuole counts (VCs) in the macrophages of the fish in the overfed group at weeks 2 and 8 were significantly greater than those of fish in the control group (Figure 3).

### 3.4. Gene Expression

The expression of ten genes (IL-1β, IL-10, CK-1, TNF-α, TLR2, CD36, SAA, CD3, HSP70, and slgIgM) were profiled in the head kidney of overfeeding-stressed fish. Among them, only HSP70 was significantly changed, with a 2-fold increase at 2 weeks (Figure 4A). The expression of IL-10, TLR2, HSP70, and CD36 was approximately 23.8, 20.1, 10.6, and 50.1 times higher in the overfed group than in the control group (Figure 4B). The expression of four genes in overfed trout was significantly higher than that in control fish.

### 3.5. Flow Cytometry Analysis

After treating trout leukocytes with OxLDL, only the P9 population (macrophage + granulocyte), not the P11 population (lymphocyte population), was significantly different in FL3-H fluorescence intensity (Figure 5). Although the FL3-H intensity was not changed in either control and OxLDL-treated cells at 6 hpi, the intensity in OxLDL-treated at 6 hpi was dramatically increased compared to the control. In the double-staining assay, the cells in the P11 population were dramatically changed according to the dose of OxLDL used to treat the trout leukocytes. More than 80% of the control lymphocytes were classified in the annexin-V and PI-negative (Anx^−^/PI^−^) sector at 48 hpi. In contrast, only 68.4, 62.7, and 52.7% of lymphocytes cultured with 0.2, 1, and 2 mg/mL OxLDL for 48 h were considered live cells (Anx^−^/PI^−^ sector), and the proportion of Anx^+^/PI^−^, Anx^−^/PI^+^, and Anx^+^/PI^+^ cells was increased as the OxLDL concentration was treated. In detail, at 24 hpi, 9.7, 14.6, and 17.4% of the early apoptotic cells (Anx V^+^/PI^−^) were found in the 0.2, 1, and 2 mg/mL OxLDL-treated groups, respectively, which were 2.5–4.5 times higher than those in the control group. At 48 hpi, late (Anx^+^/PI^+^) as well as early apoptotic cells (Anx V^+^/PI^−^) were dose-dependently increased (Figure 6).

## 4. Discussion

In this study, checking the exact feeding rate is crucial for determining the effect of overfeeding in fish. Thus, we used floating dry pellets (containing approximately 10% crude fat) only available for rainbow trout in the domestic market since we could easily collect the uneaten feed and measure its weight. In general, feed for rainbow trout is commercially manufactured to contain approximately 7–20% crude fat. Yamamoto et al. [24] reported an optimal crude fat composition for rainbow trout was 11%, although they used a little smaller size trout (approximately 30–80 g) than fish (average weight = 87 g) used in this study. Rainbow trout feed has undergone a shift since the 1970s when it was relatively high in protein and low in lipid content, to being lower in protein and higher in lipids [25]. A 10% total lipid content is commonly used in rainbow trout feed in experimental settings, which meant that this study used the nutritionally well-balanced feed [26]. Fish in the control group were fed 1% body weight per day, following the protocol for feeding experiments in trout and salmon [27]. Those authors showed that the optimal feeding rate for 8-9-inch trout at 9–15 °C was 1–1.5% of their bodyweight per day. Given that rainbow trout in the overfeeding group were fed at 3.44% of their bodyweight on average, which is approximately 2–3 times higher than recommendations, the experimental design used in this study was appropriate for determining the differences between fish in the normal fed and overfed groups.

Indeed, the difference between the feed intake of the control and overfed groups was approximately 2.5%, and this proportion of diet was quite similar to previous studies on overfeeding stress in teleost [2,28,29]. We successfully observed BMI changes as well as high lipid accumulation in the liver. These results indicated that we successfully induced overfeeding obesity in rainbow trout. Likewise, overfeeding-induced zebrafish in a previous study [30] showed significantly higher BMIs (approximately 0.07 g/cm^2^) than those of the control fish (approximately 0.04–0.05 g/cm^2^) at week 8. It is obvious that overfeeding in teleost increased BMI, but the normal BMI value can differ, depending on the species.

With the exception of the TP value in week 2, there were no significant differences in hemato-serological parameters between the control and overfeeding groups. It can be presumed that the elevated TP in overfed fish was due to a disorder in protein metabolism, immunological inflammation, and chronic inflammation due to excessive-feeding stress [31,32,33]. ALP in the overfed group at week 8 was significantly elevated over that of the control group. It is known that the major function of human ALP secreted from various organs, such as liver, kidney, bones, and adipose tissues is to prevent intracellular fat deposition and maintain cellular balance [34,35]. Several studies [34,35,36] found that higher ALP levels were linked to obesity and body fat mass in humans. Rohmah et al. [37] showed that sea-reared trout fed a high-fat diet had high ALP levels. Therefore, taken together with other relevant studies, total protein, alkaline phosphate, and hematocrit might be good indicators for overfed or obese fish.

In this study, the HSI of fish in the overfed group at weeks 2 and 8 was significantly higher than that of fish in the control group. This is important evidence that lipid accumulation in the liver occurred in overfed fish, as previously described [18,38,39]. Based on histological analysis, the vacuolated cells in the liver of the overfed fish harbored larger lipid deposits in the cytoplasm compared to fish in the control group. Vacuolated areas in the cytoplasm of hepatocytes are considered to be cytoplasmic fat vacuoles, and a large vacuolated area or higher hepatocyte cellular volume was found in the overfed fish at weeks 2 and 8. It is known that increased fat components caused by excessive nutrient uptake can trigger a cascade of hepatocellular vacuolation and enlargement of the liver in fish and mammals [2,18,40]. In particular, Taddese et al. [2] showed that fish overfed for 3 weeks harbored much higher hepatocyte volumes compared to control fish. Other studies [41,42,43] have also supported an increase in adipose tissue accumulation by surplus nutrients, which is a characteristic of obese humans and crucial evidence for the overabundance of lipids from overfeeding-induced obesity.

The expression level of several genes related to inflammation and stress was determined to investigate the biological responses in fish during overfeeding in this study. The expression levels of IL-10, TLR2, CD36, and HSP70 in fish in the overfed group at week 8 were significantly upregulated, whereas, only HSP70 was significantly higher in overfed fish at week 2. Elevations in free fatty acids in humans are known to induce oxidative stress and pro-inflammatory responses [44]. Obesity is associated with low-grade systemic chronic inflammation, which is characterized by the abnormal production of pro- and anti-inflammatory adipocytokines [45]. This group found that obese and overweight individuals showed significantly increased expression of TLR2 in both peripheral blood mononuclear cells and adipose tissue. Other studies [46,47,48] described that lipids and non-essential free fatty acids stimulated pro-inflammatory pathways, such as IL-1β, IL-6, and TNF-α, through TLR2 activation. Likewise, in this study, the expression of TLR2 in overfed fish at week 8 was significantly higher than that of control fish (*p* = 0.0023), indicating the involvement of this pattern recognition receptor during overfeeding in fish. However, unlike previous studies [45,48], there were no significant differences in the expression of pro-inflammatory cytokines, such as IL-1 and TNF-α, between the control and overfed groups during the experimental period. It is well known that IL-10 is an anti-inflammatory cytokine, limiting the production of pro-inflammatory cytokines such as IL-1β and TNF-α [49]. In this study, the expression of IL-10 in fish in the overfed group was significantly higher than that of control fish. This might be a reason why, in this study, pro-inflammatory cytokines were not upregulated in the overfed fish. CD36, one of the fatty acid transporters, was significantly upregulated in fish in the overfed group and is involved in facilitating cellular long-chain fatty acid uptake, inflammatory response, intestinal fat absorption, lipid storage in adipose tissue, and metabolic disorders, such as obesity [50,51]. Indeed, many reports have already demonstrated that the functions of CD36 in Atlantic salmon, rainbow trout, zebrafish, and carp are similar to those in mammals [52,53,54]. Surplus lipid and excessive saturated fatty acids can generate CD36-TLR2-dependent apoptosis through high CD36 and TLR2 expression, which would be a significant stressor to trigger physiological disorders [55]. A high level of HSP70, a well-known biomarker for endoplasmic reticulum stress, supported the adverse effects of overfeeding-induced obesity [56,57].

According to Haka et al. [58], adipose tissue macrophages (ATMs) help in maintaining tissue homeostasis by phagocytizing the dead adipocytes in adipose tissue. When the volume of adipocytes was inordinately increased due to obesity, ATMs phagocyte dead adipocytes resulting in hundred times larger than normal cell size [58]. However, if phagocytosis is not enough to handle all dead adipocytes, ATMs will release extracellular enzymes, such as acidic lysosomal enzymes via exocytosis to destroy the remaining dead adipocytes. As a result, the lysosomal enzymes liberate free fatty acids [58,59]. This kind of immunological interplay triggered chronic low-grade inflammation and overfeeding-induced inflammation in teleost [28]. To modulate higher pro-inflammatory cytokines (TNF-α, IL-1β, and IL-6), the host stimulated anti-inflammatory macrophages (M2), which induced anti-inflammatory cytokines (e.g., IL-10, IL-1 receptor antagonist) [60]. Moreover, providing abnormal cholesterol in feed led to higher mortality in fish species infected with *A. hydrophila* [19]. Taken together, overfeeding-induced obesity in rainbow trout can lead to lipid and cholesterol accumulation, as well as disorders of immunological modulation.

When OxLDL treatment was given and overfeeding-induced obesity was triggered in trout leukocytes, numerous lipids accumulated in the macrophages. Since overfeeding stress is highly linked to the occurrence of obesity, it is highly likely that fish in the overfed group was overweight and obesity status [3]. The fish in the overfed group harbored much more enlarged adipocytes than the control fish as they had been phagocytosing more free fatty acid and dead adipocytes, leading to more vacuoles in the cytoplasm. Indeed, macrophages regularly phagocytose dead adipocytes [59]. In some previous studies [20,58], high lipid accumulation, or loaded macrophages were termed foam cells. Foam cells are macrophages transformed by engulfing a large amount of cholesterol and lipid, which might be linked to a low-grade inflammation state, attenuated macrophage phagocytosis and pathogenesis, such as atherosclerosis and cardiovascular event [10,61,62]. Likewise, the macrophages observed in the overfed fish in this study were morphologically highly similar to foam cells found in goldfish (*Carassius auratus*) fed high cholesterol feed [20]. This indicates that overfeeding or excessive nutrients in the teleost could cause morphological changes in macrophages.

Obesity is widely known as a major factor in the formation of OxLDL, which could affect systemic metabolism and physiological homeostasis [63,64]. When the leukocytes of normal rainbow trout in this study were treated with OxLDL, the Nile Red intensity was significantly increased in a macrophage-like cell population within 6 h. The higher FL-3 intensity in the OxLDL-treated group indicated that the macrophages were capable of phagocytosing and absorbing lipids within a short time. However, no significant differences were found in the lymphocytes (data not shown). These results indicate that macrophages in teleost could also take in and process surplus lipids [20]. Furthermore, as time went by, the lymphocytes treated with OxLDL moved from the Anx V^−^/PI^−^ to the Anx V^+^/PI^−^ and Anx V^+^/PI^+^ sectors. In general, when cells initiate apoptosis, phosphatidylserine is translocated from the inner plasma membrane to the outer membrane. Translocating phosphatidylserine, which can be detected by Annexin V staining, is important evidence of early apoptosis. Hence, Anx V^−^/PI^−^, Anx V^+^/PI^−^, and Anx V^+^/PI^+^ cells have been considered live, early apoptotic, and late apoptotic cells, respectively [21,65]. In this study, although there were no significant differences between the control group and groups treated with 0.2, 1, and 2 mg/mL OxLDL until 8 hpi, the cell distribution in the OxLDL-treated group at 24 and 48 hpi was different from that of the control group. The cell distribution in the early apoptotic (Anx^+^/PI^−^) and late apoptotic (Anx^+^/PI^+^) sectors was increased according to the treatment time and dose of OxLDL. Although B lymphocytes produce antibodies against LDL and T lymphocytes can process OxLDL [66,67], high concentrations of OxLDL can cause oxidative stress, and thus apoptosis, through the Jun kinase (JUN) and mitogen-activated protein kinase (MAPK) pathways [66]. In this study, we did not investigate the pathways involved in apoptosis, but it seems that the trout lymphocytes were also significantly stressed by OxLDL. Similarly, OxLDL, which is known as a by-product of obesity, is highly involved in the damage and apoptosis of lymphocytes and macrophages, which could cause immunosuppression [66,68]. Several studies demonstrated that OxLDL triggered apoptosis, especially in CD8+ lymphocytes (T. lymphocyte) in humans [66,67,69]. However, in this study, we did not find apoptosis in the macrophages (data not shown).

## 5. Conclusions

In conclusion, in this study, overfeeding-induced obesity occurred in rainbow trout in the overfed group and showed significantly higher BMIs and HSI than fish in the control group. There were clear immuno-physiological imbalances between the two feeding groups, as seen by higher TP and ALP levels and fat deposition in the hepatocytes and macrophages, leading to the formation of foam cells and fatty changes (Figure 7). In addition, we observed apoptosis in lymphocytes exposed to OxLDL. Since apoptosis is related to the TLR2-CD36 apoptotic pathway, this indicates that an excessive amount of free fatty acid in fish might induce immunological problems (Figure 7). This study demonstrated that overfeeding fish caused severe stress and immunosuppression, which might lead to increased vulnerability to infectious diseases. The study also emphasizes how important a feeding regime and appropriate feed quantity are to maintaining fish health.

## Figures and Tables

**Figure 1 animals-10-01499-f001:**
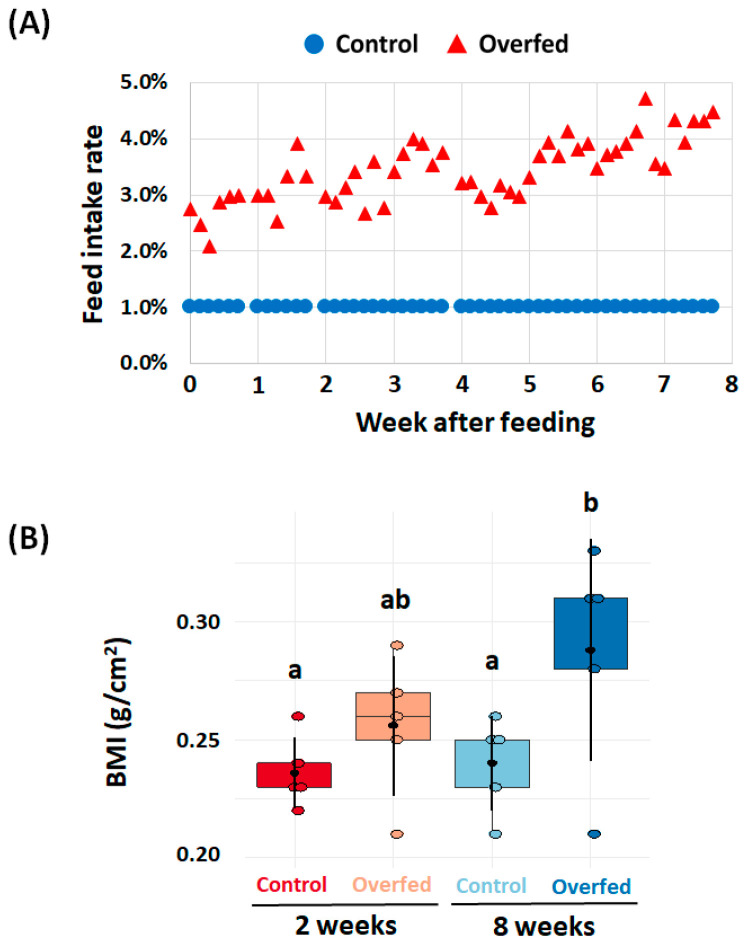
(**A**) Daily feed intake rate for 8 weeks in the control (1% bodyweight of dry commercial feed) and overfed groups. (**B**) BMI of the control and overfed trout at weeks 2 and 8. The feed intake ratio was calculated by dividing the feed weight by the bodyweight. Different letters indicate statistically significant differences determined by Duncan’s multiple range test (*p* < 0.05).

**Figure 2 animals-10-01499-f002:**
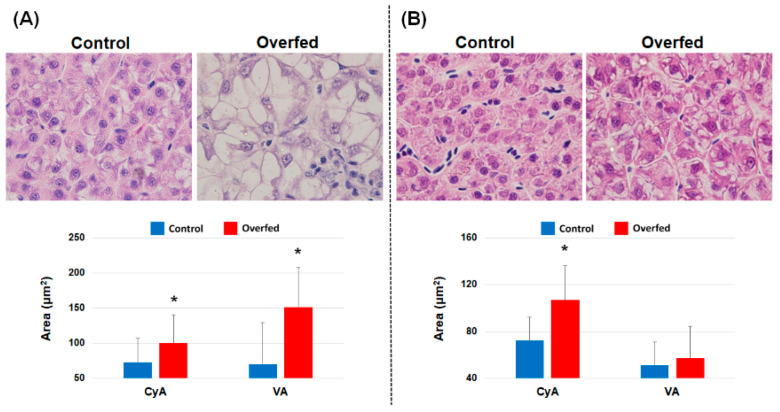
Liver histology results. Panels (**A**,**B**) represent samples taken at weeks 2 and 8, respectively. The bar plots indicate the cytoplasmic area (CyA) and vacuolated area (VA) in the hepatocytes. The results are presented as the mean ± S.D. * indicates a significant difference (*p* < 0.05 based on student’s *t*-test).

**Figure 3 animals-10-01499-f003:**
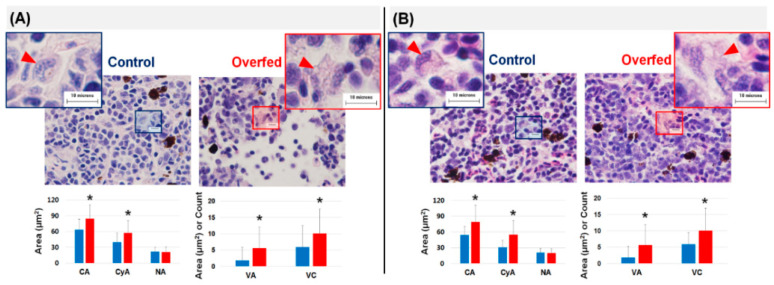
The morphological characteristics of macrophages in head kidney based on histological results. The red triangles indicate macrophages. The leukocytes in overfed trout contained much larger cytoplasmic fat vacuoles with ambiguous boundaries between the cytoplasm and extracellular space. Panels (**A**,**B**) show samples taken at weeks 2 and 8, respectively. The total cell area (CA), cytoplasmic area (CyA), nuclear area (NA), vacuolation area (VA), and vacuole count (VC) in the macrophages are presented in bar plots. The blue and red bars indicate fish in the control and overfed groups, respectively. The results are presented as mean ± S.D. * indicates a significant difference (*p* < 0.05 based on student’s *t*-test).

**Figure 4 animals-10-01499-f004:**
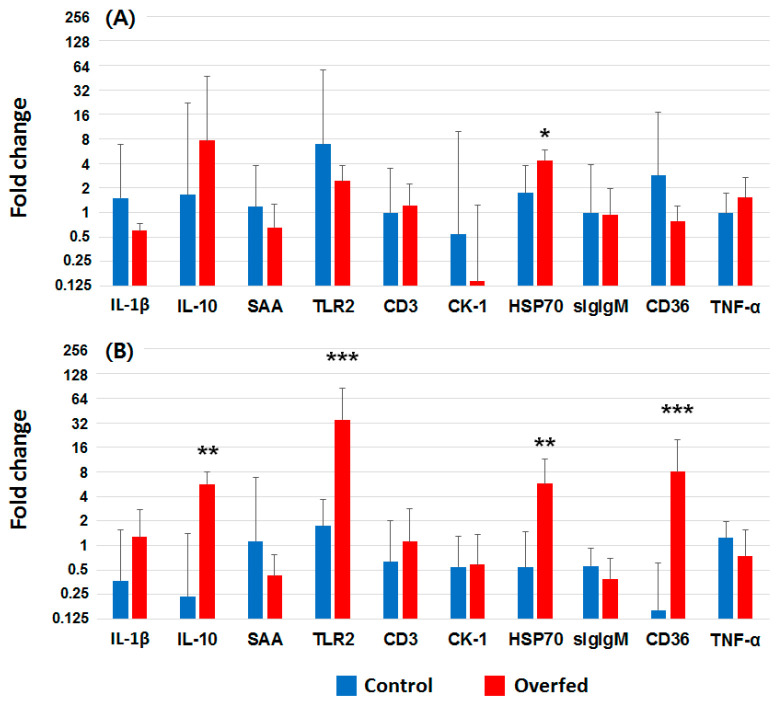
The relative interleukin-1β (IL-1β), interleukin-10 (IL-10), serum amyloid A (SAA), toll-like receptor 2 (TLR2), cluster of differentiation 3 (CD3), chemokine-1 (CK-1), heat shock protein 70 (HSP70), secreted type of IgM (sIgM), cluster of differentiation 36 (CD36), and TNF-α expression in head kidney at (**A**) 2 weeks and (**B**) 8 weeks. The results are presented as the mean ± S.D. * *p* < 0.05; ** *p* < 0.01; and *** *p* < 0.001 indicate significant differences based on student’s *t*-test.

**Figure 5 animals-10-01499-f005:**
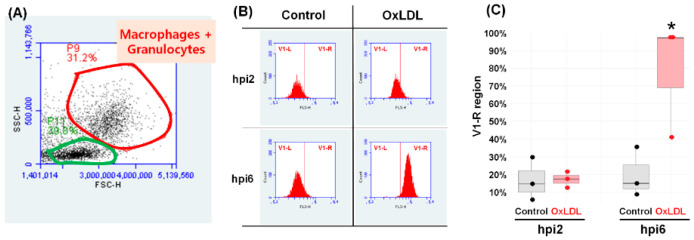
Scattergram of side scatter (SSC) plotted against forward scatter (FSC) of leukocytes isolated from the head kidney. Panel (**A**) represents the P 9 region (red line) and the P 11 region (green line), indicating macrophage and lymphocyte populations, respectively. (**B**) Nile-Red fluorescence (FL-3 fluorescence) of the macrophage and granulocytes (P9-gated cells) in the control and OxLDL group at 2 and 6 hpi. (**C**) The percent of macrophages and granulocytes (P9-gated cells) in the control and OxLDL groups with high Nile Red affinity (V1-R region). The results are presented as the mean ± S.D. * indicates a significant difference of *p* < 0.05 based on the student’s *t*-test.

**Figure 6 animals-10-01499-f006:**
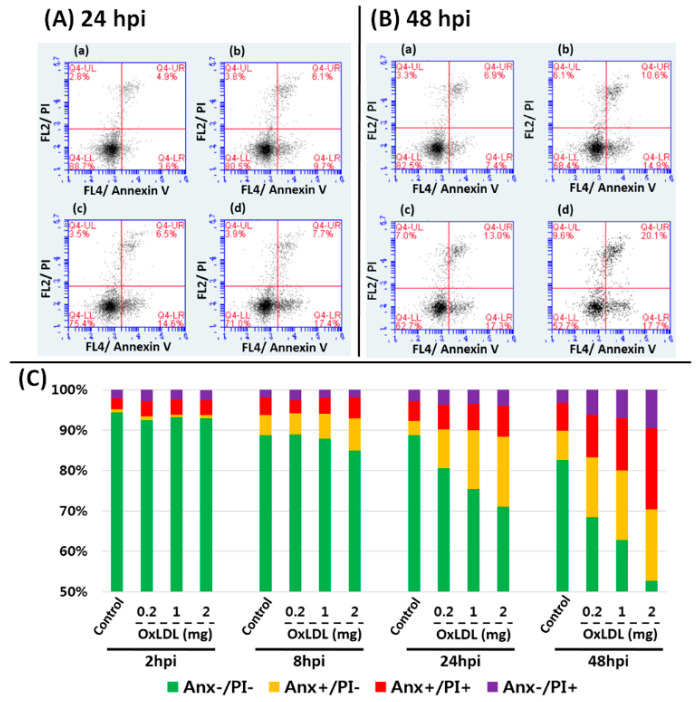
The scattergrams of FL-2 (PI) plotted against FL-4 (Anx V) fluorescence of the leukocytes under (**A**(**a**)) and (**B**(**a**)) 0 (control); (**A**(**b**)) and (**B**(**b**)) 0.2; (**A**(**c**)) and (**B**(**c**)) 1, and (**A**(**d**)) and (**B**(**d**)) 2 mg/mL OxLDL treatment at 24 and 48 hpi, respectively. (**C**) The percentage of Anx V^−^/PI^−^, Anx V^+^/PI^−^, Anx V^+^/PI^+^, and Anx V^−^/PI^+^, at 2, 8, 24, and 48 hpi are indicated in the cumulative bar graph.

**Figure 7 animals-10-01499-f007:**
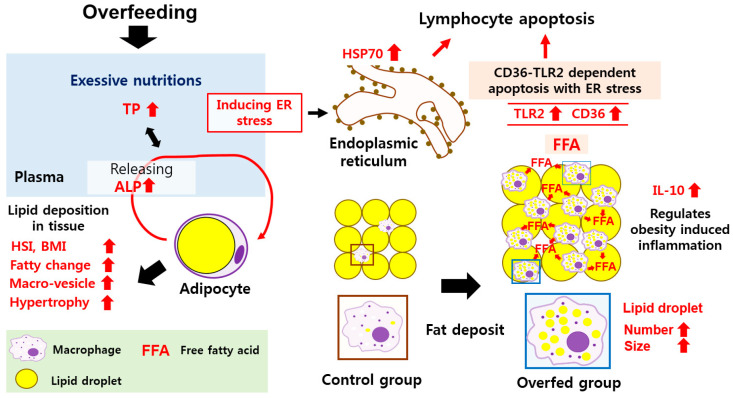
Schematic representation of the biological response of overfed rainbow trout.

**Table 1 animals-10-01499-t001:** Hepatosomatic index (HSI) and serological/hematological changes (mean ± S.D.) in rainbow trout overfed for 2 and 8 weeks.

Parameters ^+^	At Week 2		At Week 8	
	Control	Overfed	Control	Overfed
HSI (%)	0.93 ±0.15	1.69 ± 0.4 **	0.83 ± 0.11	1.43 ± 0.57 *
RBC (10^8^ cells/mL)	10.8 ± 3.86	12 ± 4.51	10.7 ± 1.56	10.5 ± 1.94
Hematocrit (%)	36.6 ± 2.1	41.4 ± 4.04	33.4 ± 1.8	42.6 ± 2.88 ***
Hemoglobin (g/dL)	5.82 ± 1.24	7.55 ± 2.37	5.71 ± 0.84	6.60 ± 1.64
MCV (μm^2^)	328 ± 71	376 ± 103	319 ± 55	415 ± 71
MHC (pg)	55 ± 14	64 ± 10	54 ± 9	65 ± 23
MCHC (%)	17 ± 3	18 ± 5	17 ± 2	15 ± 3
TP (g/dL)	4.5 ± 0.4	5.3 ± 0.6 *	4.9 ± 0.4	5.9 ± 0.9 *
ALP (U/L)	603 ± 238	767 ± 257	665 ± 251	991 ± 166 *
GOT (U/L)	409.2 ± 152.5	364 ± 54	440.4 ± 128.3	406 ± 73
GPT (U/L)	14.6 ± 8.6	13.80 ± 2.2	20.8 ± 9.4	15.8 ± 2.1
GLU (mg/dL)	131.8 ± 15.7	118.8 ± 3.4	112.2 ± 12.5	113.6 ± 20.3
TCHO (mg/dL)	359.0 ± 37.3	394.2 ± 39.0	442.8 ± 9.7	465.4 ± 50.5
LDH (U/L)	2268 ± 1200	1397 ± 1127	820.8 ± 481.4	846 ± 139
Ca (mg/dL)	12.4 ± 0.5	11.9 ± 0.65	9.4 ± 0.4	9.44 ± 0.36

* Significant difference, *p* < 0.05; ** Significant difference, *p* < 0.01; *** Significant difference, *p* < 0.001 based on student’s *t*-test. ^+^ RBC (red blood cell), MCV (mean corpuscular volume), MHC (mean corpuscular hemoglobin), MCHC (mean corpuscular hemoglobin concentration), HSI (hepatosomatic index), TP (total Protein), ALP (alkaline phosphatase), GOT (glutamic oxaloacetic transaminase), GPT (glutamate pyruvate transaminase), GLU (glucose), TCHO (total cholesterol), LDH (lactate dehydrogenase), and Ca (calcium).

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
