# Peer review of "Overfeeding-Induced Obesity Could Cause Potential Immuno-Physiological Disorders in Rainbow Trout (Oncorhynchus mykiss)"

_animals, 2020, doi:10.3390/ani10091499_

Round 1
Reviewer 1 Report
The manuscript is interesting and well presented. There are major points to be addressed; please see the attached document for the detailed review. The authors should consider better integrate the manuscript to the scope of the Special Issue by contextualizing the nutrition research and fish farming practices about overfeeding and its implication to animal physiological status, welfare and production.

Author Response
Dear reviewer1
Thank you for your review and very promising comments. They are very helpful to improve the manuscript. Please check our responses and the revised manuscript.
Reviewer 1.
The manuscript is interesting and well presented. There are major points to be addressed; please see the attached document for the detailed review. The authors should consider better integrate the manuscript to the scope of the Special Issue by contextualizing the nutrition research and fish farming practices about overfeeding and its implication to animal physiological status, welfare and production.
Line 26 – 27: It would be interesting to add a sentence to integrate these findings to their application and impact in aquaculture research and industry.
- A sentence has been added in L 26-29.
Line 36. He (??)
-It was deleted.
Introduction: The first paragraph is interesting but it would be beneficial to add a sentence contextualizing why fish farms overfeed? Is it to maximize growth and reduce production cycle? It would be also beneficial the authors to add their definition of overfeeding. is the a threshold that indicates overfeeding? For example 6% biomass? Also, the special issue is about modern formulations. I recommend the authors to link modern formulations with overfeeding. The last paragraph briefly mentioned that, but it should be expanded.
There are several fish farming manuals for important aquaculture species including rainbow trout (e.g., one from FAO and Leitritz). According to standards, however, different amounts of feed are supposed to be provided to fish, depending on their size and water temperature. This makes difficult for farmers to provide recommended amounts, and thus they usually tend to feed until they refuse (overfeeding). In general, given that optimal feeding rate for 8 – 9 inch trout when acclimated at 9 – 15 °C is 1 – 1.5% of fish body weight, providing 3.4% of fish body weight is more than two times higher than recommendation. Some sentence in the first paragraph is revised in line 57 – 59.
Line 76 – 77. I could not understand this sentence. Please revise
- We revised this sentence in line 82 - 84.
Line 86 – 91: The authors need to add more information about the experimental system was it flow through or recirculation aquaculture system. What was the water quality? What was the volume of the tanks?
- We added much more detail information in material and methods section (Line 93 – 94, 96).
Line 90. replace "crude" with "total"
- We used the same word in accordance with manufacturer`s suggestion.
Line 94. The authors need to expand that adding more details such as:
- feeding frequency - hand-fed or use autofeeders
how long did each feeding event last? / It is important to differentiate overfeed to feeding to apparent satiation.
- The feed at least three times a day by hand. This information is added in line 101 - 104.
Line 157 – 158. Please add which test(s) was/were used for the ANOVA assumptions.
- Thank you for pointing out. We clarified them in line 167 - 170.
Line 163. Group -> Groups
- Thank you for your correction.
Line 167. Past verb tense – had
- Thank you for your correction.
Figure 1. It seems the authors use the average data to plot the figures. It would be interesting to plot each replicate to understand any variation within treatment. Please justify the reason why average data was used.
- We used two big tanks, control and overfed group, respectively. We had not replicate tanks within a treatment.
Line 261 – 262: I could not understand the context of which this sentence refers to. Please revises
- This sentence was revised in line 270 - 271.
Line 265 – 266: Please add a reference. I believe the common knowledge is that nutritional demands and requirements are higher in smaller animals compared to larger fish.
- In general, the percentage of lipids in commercial feed composition is higher for heavier trout. We deleted it, because the sentence seems to be not highly related to this study.
Line 268 – 270: Please the values to illustrate what the authors mean by high and low protein and lipid content.
- We empathized the feed used in this study was appropriate. We illustrated it in detail.
Line 273 – 275: It would be beneficial here to point out the difference between overfeeding and feeding to apparent satiation.
- This explanation was added in line 282 - 284.
Line 278 – 280: This sentence is not clear. I could not fully understand
- We rearranged the first paragraph of discussion section, and this sentence was deleted.
Line 282 – 286: It would be interesting to contextualize what the most nutrition research and the industry uses as a feeding approach. Do they overfeed the animals?
- In the industrial fields, especially fish farms, providing an excessive amount of feed has occasionally happened. But, it is not a common problem for nutrition research.
Line 363 – 364: The authors need to better link obesity to the current scope of the study.
- We added an additional sentence to the link between overfeeding stress and obesity in line 359 - 362. In addition, we changed the overfeeding stress to overfeeding-induced obesity, because overfeeding stress is known to highly correlate with obesity.

Reviewer 2 Report
Overall, this study seems well designed and executed, the analyses are appropriate, and the conclusions well discussed. I have several specific suggestions for improvement of the manuscript below.
Line 26: remove first use of "stress"
Line 36: what is the "He" hepatosomatic index? Is this a typo?
Line 78: "overfeeding is common". By what measure? Citations?
Line 85: move Oncorhynchus mykiss to first mention of rainbow trout in introduction.
Line 94: "until they refused to eat more". The fish were fed to satiation throughout the duration of the study, correct? This is not what is indicated by the abstract, which suggests that fish were fed a specific rate of 3.44% throughout the study.
Line 96: "post overfeeding" is confusing. Usually, post "an action" is used to indicate that the action has been concluded. However, in this study, the fish are being overfed throughout the study duration, so I think what is meant by post overfeeding, is weeks after overfeeding was initiated.
Line 100: Provide equation or description of calculation for hepatosomatic index.
Lines 106 and 108-110: remove units of measure from equations. These should be put after first mention of the parameter in the first sentence.
Line 111: citation for "standard methods"?
Line 164: looks like 3.44% is an average over the course of the study (with an increasing rate overall). Please include some form of error or range for the measurements to indicate that 3.44% was not a fixed rate.
Figure 1: what is the "+" for on the y axis in panel A? Similar to comment above, x-axis appears to have an incorrect use of "post feeding". In the caption, replace "rate" with "ratio" for consistency.
Line 175-176: Entirety of first sentence can be removed as these results and the parameters measured are explained in the table. Suggest starting the paragraph with the second sentence, "Of the parameters evaluated, ...", and then reference the table at the end of the sentence.
Line 179 and throughout results: results would be more useful to the reader if instead of using "different" you provided a direction (e.g., the hematocrit and ALP were significantly higher).
Lines 178-184: Remove values presented in parentheses throughout. The values are in the table and do not need to be repeated in the text. Reference the table instead.
Line 194: Remove the first part of the sentence. Start the sentence at "Different types" and modify as needed to include information regarding the livers of overfed fish.
Figure 2: Remove "At week 2" and "At week 8" from the figure. This is in the caption.
Lines 213-214: suggest removal of entire sentence following "However, at week 8," and then follow this with the entirety of the last sentence of the paragraph.
Lines 262-263: It was never mentioned in the methods that uneaten feed was collected and weighed. This should be described in the methods, detailing how it was used and why.
Line 265: "grower fish"...never heard this term before. Please define.
Lines 339-340: confusing wording (especially multiple uses of "dead").
Line 398: "Increased vulnerability to infectious disease". This as not directly measured in this study, and not well discussed. This thought could be expanded on for clarity. By what mechanism would this occur?
Author Response
Dear reviewer 2.
Thank you for your promising and valuable comments and advice. Thanks to your considerable review, we are able to improve the manuscript. You can see our responses to your comments and the revised manuscript.
Reviewer 2.
Overall, this study seems well designed and executed, the analyses are appropriate, and the conclusions well discussed. I have several specific suggestions for improvement of the manuscript below.
Line 26: remove first use of "stress"
- We revised it.
Line 36: what is the "He" hepatosomatic index? Is this a typo?
- Thanks for your comment. It was revised.
Line 78: "overfeeding is common". By what measure? Citations?
We added a reference in the introduction (line 57 – 59).
Line 85: move Oncorhynchus mykiss to first mention of rainbow trout in introduction.
- It was moved to the first mention of rainbow trout in the introduction.
Line 94: "until they refused to eat more". The fish were fed to satiation throughout the duration of the study, correct? This is not what is indicated by the abstract, which suggests that fish were fed a specific rate of 3.44% throughout the study.
- We gave the feed to trout until they do not eat, and they ate 3.44 % of their bodyweight a day on average. This information was added to the abstract section.
Line 96: "post overfeeding" is confusing. Usually, post "an action" is used to indicate that the action has been concluded. However, in this study, the fish are being overfed throughout the study duration, so I think what is meant by post overfeeding, is weeks after overfeeding was initiated.
- Thank you for your comment “Weeks post overfeeding (wpo) was revised to “Weeks after overfeeding (wao)” throughout the paper.
Line 100: Provide equation or description of calculation for hepatosomatic index.
The calculation was described.
Lines 106 and 108-110: remove units of measure from equations. These should be put after first mention of the parameter in the first sentence.
The units of measure from equations were deleted.
Line 111: citation for "standard methods"?
This is the manufacturer`s method. “Standard methods” were changed to “manufacturer`s methods”.
Line 164: looks like 3.44% is an average over the course of the study (with an increasing rate overall). Please include some form of error or range for the measurements to indicate that 3.44% was not a fixed rate.
Thank you for your comments. The standard range of feeding rate was added in line 175.
Figure 1: what is the "+" for on the y axis in panel A? Similar to comment above, x-axis appears to have an incorrect use of "post feeding". In the caption, replace "rate" with "ratio" for consistency.
The errors of the y-axis in panel A and “post-feeding” were revised.
Line 175-176: Entirety of first sentence can be removed as these results and the parameters measured are explained in the table. Suggest starting the paragraph with the second sentence, "Of the parameters evaluated, ...", and then reference the table at the end of the sentence.
It was deleted as following your comment.
Line 179 and throughout results: results would be more useful to the reader if instead of using "different" you provided a direction (e.g., the hematocrit and ALP were significantly higher).
Thank you for your comment. The word “different” was revised to “higher”.
Lines 178-184: Remove values presented in parentheses throughout. The values are in the table and do not need to be repeated in the text. Reference the table instead.
Most values that you mentioned were removed.
Line 194: Remove the first part of the sentence. Start the sentence at "Different types" and modify as needed to include information regarding the livers of overfed fish.
The first sentence was revised as following your comment.
Figure 2: Remove "At week 2" and "At week 8" from the figure. This is in the caption.
The caption “At week 2” and “At week 8” in Figure 2 were removed.
Lines 213-214: suggest removal of entire sentence following "However, at week 8," and then follow this with the entirety of the last sentence of the paragraph.
It was revised as your comment.
Lines 262-263: It was never mentioned in the methods that uneaten feed was collected and weighed. This should be described in the methods, detailing how it was used and why.
Uneaten feeds were collected within 30 min after providing to trout, and they were not counted for feed intake rate. This information was added in 2.2 paragraph.
Line 265: "grower fish"...never heard this term before. Please define.
This sentence was removed.
Lines 339-340: confusing wording (especially multiple uses of "dead").
This sentence was revised.
Line 398: "Increased vulnerability to infectious disease". This as not directly measured in this study, and not well discussed. This thought could be expanded on for clarity. By what mechanism would this occur?
We found that overfeeding-induced obesity makes the abnormal expression of immuno-physiological biomarkers such as IL-10, TLR2, HSP70, and CD36. Also, we demonstrated that OxLDL which is a by-product of overfeeding-induced obesity could trigger trout lymphocyte apoptosis. This weakness could be linked to vulnerability to infectious diseases, although further studies are necessary. We revised the sentence a little about the vulnerability of infectious diseases in line 406 and 408.
